# Nanostructured Thermoelectric Films Synthesised by Spark Ablation and Their Oxidation Behaviour

**DOI:** 10.3390/nano13111778

**Published:** 2023-05-31

**Authors:** Hendrik Joost van Ginkel, Lisa Mitterhuber, Marijn Willem van de Putte, Mark Huijben, Sten Vollebregt, Guoqi Zhang

**Affiliations:** 1Department of Microelectronics, Faculty of Electrical Engineering, Mathematics and Computer Science, Delft University of Technology, 2628 CD Delft, The Netherlands; h.j.vanginkel@tudelft.nl (H.J.v.G.); s.vollebregt@tudelft.nl (S.V.); 2Materials Center Leoben Forschung GmbH, A-8700 Leoben, Austria; lisa.mitterhuber@mcl.at; 3MESA+ Institute for Nanotechnology, University of Twente, 7522 NH Enschede, The Netherlandsm.huijben@utwente.nl (M.H.)

**Keywords:** thermoelectric, nanoparticle, Bi_2_Te_3_, spark ablation, nanostructured

## Abstract

Reducing the thermal conductivity of thermoelectric materials has been a field of intense research to improve the efficiency of thermoelectric devices. One approach is to create a nanostructured thermoelectric material that has a low thermal conductivity due to its high number of grain boundaries or voids, which scatter phonons. Here, we present a new method based on spark ablation nanoparticle generation to create nanostructured thermoelectric materials, demonstrated using Bi_2_Te_3_. The lowest achieved thermal conductivity was <0.1 W m${}^{-1}$ K${}^{-1}$ at room temperature with a mean nanoparticle size of 8±2 nm and a porosity of 44%. This is comparable to the best published nanostructured Bi_2_Te_3_ films. Oxidation is also shown to be a major issue for nanoporous materials such as the one here, illustrating the importance of immediate, air-tight packaging of such materials after synthesis and deposition.

## 1. Introduction

Thermoelectric (TE) materials are materials that produce a potential difference when exposed to a temperature gradient or vice versa. This property can be exploited using a p- and an n-type TE material in a Peltier or a Seebeck module. Peltier modules apply a potential difference across the TE material to force a heat flow against the temperature gradient, cooling one side and heating the other. Such active cooling can be applied in refrigeration or active cooling of hot surfaces [1]. Seebeck modules use the temperature difference to generate a potential difference, which results in a current [2,3,4,5]. Such modules can recover and convert waste heat generated by hot surfaces or electronic components into electrical energy, potentially boosting the efficiency of LEDs and other microelectronic devices. They can also be used as miniature power generators, called thermoelectric generators (TEGs), harvesting heat from the environment to power small devices such as pacemakers or wearable electronics [4,6]. Since TE materials are semiconductors and TE devices have no moving parts, they have straightforward integration in IC or MEMS devices.

Unfortunately, due to the low efficiency of TE materials, large-scale application is economically not yet feasible, unless costs also drastically decrease [2,6,7,8,9,10,11,12]. The exception is in such niche applications where efficiency is not the major concern (e.g., space exploration). TEGs have been used for decades by NASA and other space agencies, and likely will be for the foreseeable future [13]. TEGs could replace batteries, especially in applications where regular battery replacement is unwanted and the power requirements are low. Devices can thus be ‘made to forget’. Examples are sensors or controllers for the internet-of-things or environmental monitoring [2,6,14]. On the other end, TE-based solid-state cooling would be highly useful in cryo-TEM or quantum computing electronics [15].

As mentioned, conversion efficiencies of TE materials are still low, at only a few percent of efficiency [2,6,7,8,9,10,11,12]. To improve energy conversion efficiency, a few properties need to be optimised, which are expressed in a dimensionless figure of merit (FoM):(1)$$ZT=\frac{S^{2}\sigma T}{\kappa_{e}+\kappa_{l}}$$

Here, *S* is the Seebeck coefficient in V K${}^{-1}$, σ is the electrical conductivity in S m${}^{-1}$, *T* is the operating temperature in K, and $\kappa_{e}$ and $\kappa_{l}$ are the electrical and lattice contributions of the thermal conductivity in W m${}^{-1}$ K${}^{-1}$. $S^{2}\sigma$ is known as the power factor and is an expression of the power generation performance of the material. Since all parameters are material dependent (except for the operating temperature T), it is possible to optimise the material properties. Evidently, the ideal TE material has high electrical conductivity, a high Seebeck coefficient, and virtually no thermal conductivity. This concept is known as the phonon glass, electron crystal (PGEC) approach and was first described by Slack [16]. Although this ideal material is highly sought after, it has never been actually made. The central problem for TE material optimisation is that *S*, σ, and κ are all dependent on material properties such as carrier mobility, carrier concentration, and crystal structure [5,10,17,18,19]. Optimising one parameter will change the performance of another and can result in little-to-no improvement, or even reduction, in ZT. This trade-off is the cause of the slow progress in TE materials research over the last decades. Even today, decoupling these parameters is the main challenge in designing new TE materials.

The introduction of grain boundaries and voids to scatter phonons is one approach to reduce thermal conductivity. However, grain boundaries and voids also scatter electrons, making it a difficult trade-off. Creating nanostructured materials is a way to introduce such scattering grain boundaries, and many synthesis and nanostructuring methods have since been developed to create nanostructured TE materials [20,21,22,23,24,25,26,27,28,29].

This paper introduces a new method to produce nanostructured TE materials, demonstrated with undoped Bi_2_Te_3_. This method is based on spark ablation, a method to create nanoparticle aerosols that are deposited using an impaction printer. The result is a nanostructured thin film consisting of a network of nanoparticles with an identical composition to the parent material. Adjusting the composition of the electrodes or even using multiple nanoparticle generators together creates a wide range of potential compositions to be synthesised. In combination with a printing nozzle, it is possible to directly write nanoparticle films in complex patterns without masks or lithography. This deposition method is compatible with many substrates, including hard substrates such as silicon or metals, and softer substrates such as polymers or paper [30]. Furthermore, nanoparticle synthesis requires only a parent material to ablate and a carrier gas, making a low-waste and clean process. These advantages make spark ablation films straightforward to integrate into electronics. A first attempt to synthesise a TE film using spark ablation is presented here, without any optimisation of the material, with the goal of demonstrating the feasibility of this technology. The performance is discussed, and recommendations on how to improve the process and material are made.

## 2. Experimental

### 2.1. Nanoparticle Generation and Deposition

Spark ablation generation is the process of ablating two opposing metallic or semiconductor rods by a repeating high-voltage electrical spark to generate a supersaturated metal vapour in a carrier gas. Rapid condensation of this vapour into nanoparticles creates an aerosol of 2 to 20 nm-sized metal or metal oxide nanoparticles [30]. The synthesis and deposition process is illustrated in Figure 1. By generating several hundred sparks per second, a continuous nanoparticle flow is generated that is carried to a nozzle that accelerates the gas at hypersonic speed toward the target substrate where they land [30,31]. By moving the nozzle in three dimensions, one can directly write complex, programmable patterns without lithography. With the system used here, we could achieve a minimum line width of 120 μm, but the features presented here were all larger to simplify analysis. For even smaller features, they can be patterned using conventional lithography and lift-off procedures [32].

The setup is shown schematically in Figure 2. A spark ablation generator (VSP G1, VSParticle B.V.) was used with a pair of 99.999% Bi_2_Te_3_ electrodes supplied by VSParticle B.V. The ablation settings were 1 kV and 3 mA with 1.5 L min${}^{-1}$ of Ar with 2% H_2_ as carrier gas. The addition of hydrogen to the carrier gas is known to suppress oxidation of the metallic nanoparticles, which under normal circumstances oxidise rapidly due to their clean and unprotected surface [33,34]. The impactor used here was a prototype device developed by VSParticle B.V., which operated at 0.5 mbar with a nozzle throughput of 0.33 L min${}^{-1}$. Printing settings were, unless specified differently, 1 mm per minute with a 1 mm nozzle distance.

### 2.2. Structure Characterisation

A Bruker D8 Discover diffractometer was used for XRD measurements. Samples were prepared on a Si(100) die. An XL30 SFEG SEM from Thermo Fisher Scientific was used for SEM at 10 kV. TEM images for the size distribution were made with a JEOL 1400, operated at a 120 kV acceleration voltage. For HR-TEM, an FEI cubed Cs corrected Titan TEM was used, operated at 300 kV. The nanoparticles were deposited with a writing speed of 200 mm min${}^{-1}$ to create a film of less than a full monolayer in thickness on Si grids with 50 nm AlO${}_{x}$ membranes made in-house. Regular grids of carbon membrane on a Cu mesh would be destroyed by the force of the deposition jet and could not be used. AlO${}_{x}$ membranes are much stronger, but their transparency is not as uniform as a carbon film due to large crystal grains in the AlO${}_{x}$ layers. This made particle size analysis difficult to automate, so particles were measured manually using Fiji (v2.0.0)

The porosity was determined using a method developed by our group [35]. It is an indirect method to determine the porosity of a material by creating identical deposits and extracting their density from different measurements. It uses a quartz crystal microbalance (QCM) to measure the mass deposition rate during deposition. OpenQCM hardware with modified software was used to read out the 10 MHz quartz crystals, which gave a 6.2 ng resolution. The deposition was carried out by printing concentric circles on the QCM electrode area and was kept short to satisfy the conditions of QCM usage. Immediately afterwards, in the same deposition session, a line was printed with identical settings on a separate substrate to ensure minimal process changes. A profilometer—here, a Bruker Dektak 150—was used to measure the cross-sectional area of this line in three locations to account for local differences. At each location, the average of three measurements was used. Multiplication of the resulting area with the printing speed gave a volume deposition rate. Dividing the mass deposition rate by the volume deposition rate resulted in a density, $\rho_{film}$. This was, in turn, used to calculate the porosity $\Theta_{p}$ using the bulk porosity $\rho_{{\mathrm{Bi}}_{2}{\mathrm{Te}}_{3}}$:(2)$$\Theta_{p}=1-\rho_{film}/\rho_{{\mathrm{Bi}}_{2}{\mathrm{Te}}_{3}}$$

### 2.3. Thermoelectric Characterisation

After the initial synthesis and material analysis, its performance as a TE material had to be tested. First, the Seebeck coefficient was measured using a home-built setup shown in Figure 3. The film was printed on a 20 by 5 mm strip of silicon wafer with 100 nm thermally grown SiO_2_ and four 10/100 nm Cr/Au electrodes for electrical contact.

Electrical conductivity was measured in situ using 4-point-probe chips wirebonded to a DIL package, which was mounted on a PCB inside the printing chamber. The PCB was connected to a Keithley 2450 sourcemeter and set at a constant value of 10 mA during readout. During deposition, the nozzle moved over the four electrodes, while the sourcemeter read the resistance, creating a conducting line. For the long-term degradation study, measurements were performed on a Cascade Microtech microprobestation at ambient conditions.

The thermal conductivity of the films was investigated using time-domain thermoreflectance (TDTR) with the Netzsch PicoTR instrument. The measurement parameters of the pump–probe technique can be found in [1]. The samples were measured by using the principle of the bidirectional heat flow approach [36,37]. For doing so, the films of interest with a thickness of >15 μm were deposited on 1 mm-thick quartz with a 100 nm Al layer on top. Both lasers were focused at the interface between Al and glass, as they pass through the glass substrate according to their wavelengths of 775 nm (probe) and 1550 nm (pump). The Al layer acted here as a reflection layer for the probing and an absorption layer to guarantee uniform heating. The properties of the Bi_2_Te_3_ were deduced from the cooling curve. Therefore, an analytical model of the cooling was set up that included two heat flows. The first was the interface conductance between Al and glass, as well as through the glass substrate. The second one was the heat flow through the Al layer, the interface conductance between Al and through the Bi_2_Te_3_, and the Bi_2_Te_3_ layer. For the analysis of the film, its density of 3.4 kg m${}^{-3}$ and its specific heat of 127 J mol${}^{-1}$ K${}^{-1}$ served as input parameters for the evaluation. Furthermore, the thermal conductivity values of the Al layer, the glass substrate, and the interface conductance between Al and glass were obtained by performing TDTR measurements on the sample without the Bi_2_Te_3_ films. Because of logistical difficulty in shipping the samples for TDTR measurement, it was not possible to measure the thermal conductivity earlier than presented.

## 3. Results and Discussion

### 3.1. Nanostructure and Composition

Figure 4 shows that the deposited material was pure polycrystalline Bi_2_Te_3_; no other phases were detectable. The grain size could be extracted from an XRD spectrum with the Scherrer equation, which was applied to the highest peak. The full calculation and results are enclosed in Section A.1. The fresh sample had an approximate grain size of 5.1 nm, while the week-old and month-old samples were slightly larger, at 5.3 and 5.5 nm. An increase that small can be attributed to measurement error, such as a slightly different measurement location or orientation of the sample. It can therefore be concluded there was no significant change in the Bi_2_Te_3_ grain size over time. Although it is not accurate to attribute all peak widening to grain size and the calculation assumes perfectly spherical particles, we can confidently say that the grain size was approximately 5 to 6 nm and that there were no large particles in the samples [38]. The existence of a crystalline Bi_2_Te_3_ phase even after 1 month shows that the samples did not degrade significantly within this timespan.

TEM images, such as the ones in Figure 5, showed agglomerated nanoparticles with extensive neck formation or even fusion of particles. The images indicate that the surface of the nanoparticles consisted of highly mobile atoms that fused easily. The size distribution in Figure 6 shows a log-normal distribution, as typically found in spark ablation generation [30,38]. The mean particle size of 8±2 nm was slightly larger than the mean found by XRD. Two effects can contribute to this: a non-crystalline Bi_2_Te_3_ outer layer or an amorphous oxide shell. In both cases, XRD would only work on the crystalline core, ignoring the outer layer.

A side view of a printed film, as shown in Figure 7A, showed a porous but continuous film, similar to compacted powders. The film was 2.1 μm thick and had a high surface roughness, visible in Figure 7B.

The mass deposition process can be seen in Figure 8A and showed a similar characteristic apparent mass deposition rate as the one described in Van Ginkel et al. [35]. This was due to the decreasing mass sensitivity near the edges of the electrode, resulting in a lower response of the QCM for the same deposition rate. Therefore, the beginning and end points were the only relevant points to determine the total mass change during deposition accurately. As seen in Figure 8B, after reaching a mass of 18.60 μg, the deposit mass rapidly increased and stabilised under the protective Ar/H_2_ atmosphere at around 19.25 μg. When the gas flow was stopped, the chamber was pumped down, and leaked-in oxygen could react with the deposit, rapidly increasing the reaction rate. However, since the mass increased immediately after deposition, we must conclude that even the 99.999% pure Ar/H_2_ gas contained enough impurities to begin oxidation and, thus, oxidation must have started during deposition.

The experiment was stopped at 20.45 μg, a total mass increase of 9.9%. Total oxidation of the sample, consisting of pure Bi_2_O_3_ and TeO_2_, would have increased the mass by 18%, from 800.74 u to 944.76 u [39,40]. Bando et al. (2000) also found that, even after long term exposure to air at room temperature, the oxide layer does not exceed 2 nm, even after 5700 h [40,41]. In contrast to their experiment on a solid monocrystalline Bi_2_Te_3_, the nanoparticles here were in contact with oxygen residuals in all directions. Yet, the oxidation reaction appeared to still be slow. Oxidation was not finished, but all mass increase could not solely be attributed to oxidation, particularly after closing the clean gas flow over the sample. Water or organic compounds in the air could have adsorbed on the surface, which, due to the high surface area of the nanoporous film, could have been in significant quantities. This was, however, not visible on the SEM or TEM images, since they formed a thin film and were volatile, desorbing in high vacuum systems and under electron beam irradiation.

Using three QCMs, a mean deposition rate of $2.93\times 10^{-3}\pm 4.81\times 10^{-4}$ mg min${}^{-1}$ was measured. Combining this result with a one-minute deposition volume of $8.89\times 10^{-7}$ cm${}^{3}$ gave a mean density of 3.40±0.33 g cm${}^{-3}$, or a 0.44±0.04 porosity. This is significantly lower than previously found for Au [35] due to the lower malleability of Bi_2_Te_3_, leading to less restructuring upon impact during deposition. It was also previously found that the density of Au films is insensitive to variation in the synthesis parameters, as tested by van Ginkel et al., giving a constant value for all syntheses within the operating conditions of the setup [35]. It is therefore assumed that all samples here had the same density (and porosity) too, because the synthesis conditions were nearly identical.

### 3.2. Thermoelectric Performance

The key metric for a TE material is its figure of merit ZT (Equation (Equation 1)), which cannot be measured directly, but is calculated from several measurements. A negative Seebeck coefficient of −88.3±1.2
μV K${}^{-1}$ was obtained (see Figure 9), indicating that we had n-type Bi_2_Te_3_. Bulk Bi_2_Te_3_ grown from a melt is, by its nature, usually p-type, while n-type Bi_2_Te_3_ is made by either doping with, for example, I or Br, or by having an excess of Te [12]. Most nanostructured Bi_2_Te_3_ is also n-type, possibly due to vacancies or surface states that act as dopants [24,29,42,43]. Interestingly, ageing the sample in the air seemed to improve the Seebeck coefficient, which stabilised at −105.1±1.6
μV K${}^{-1}$. This value is lower than most reported values for Bi_2_Te_3_, at −100 to −250 μV K${}^{-1}$, but is not unusual for low conductivity samples [8,24,44,45,46]. It must be noted that the Bi_2_Te_3_ used here was undoped, and no attempts were made to optimise the material’s Seebeck coefficient.

The electric conductivity was measured during deposition in vacuum on packaged and wirebonded samples. The four electrode devices could only create a measurement when the nozzle had deposited on all four electrodes, after which an initial resistance was measured and a sharp drop in resistance occurred while the nozzle moved over the last electrode. We can see this exact behaviour in Figure 10, with a resistance minimum of 38.82Ω. After deposition, there was an immediate increase in resistance that slowed down gradually, but did not come to a complete halt. After 25 min, the resistance already increased by 34% and measurements using a four-point-probe system, as seen in Figure 11, showed that the degradation in the first week was fast, but levelled off after this. However, the conductivity did reduce further and did not seem to have reached the minimum, even after 56 days. These results show that it is critical that a deposit is protected immediately after deposition, or the electrical performance will deteriorate rapidly.

As introduced earlier, the primary aim of the experiments presented in this paper was to reduce the thermal conductivity of the Bi_2_Te_3_ by nanostructuring and improving the figure of merit (Equation (Equation 1)). Figure 12 shows the thermal conductivity of three samples (S1 to S3). S1 and S2 had identical settings (1 kV, 3 mA, 1.5 L min${}^{-1}$ Ar/5% H_2_), while S3 was made with a spark at 0.8 kV. All samples showed low thermal conductivity <0.4 W m${}^{-1}$ K${}^{-1}$, with the lowest value well below 0.2 W m${}^{-1}$ K${}^{-1}$, comparable to the lowest values reported in the literature, and a nearly tenfold reduction compared to the bulk thermal conductivity [8]. This reduction can be attributed to the nanostructure of the film, introducing grains in the <10 nm range where phonons are scattered at the grain boundaries, as demonstrated before for Bi_2_Te_3_ [20,42,44,47,48,49,50]. The porosity introduces further boundary scattering of the phonons in addition to the grain boundaries [42,48]. The 0.8 kV sample was expected have a slightly smaller mean particle size, which would have contributed to a lower κ. However, the value spread between the three samples made it impossible to attribute the lower value to a smaller particle size alone.

One month later, a further reduction in κ could be observed. This was likely due to degradation of the film by oxidation. An oxide shell around the nanoparticles will thermally insulate the nanoparticles. The samples were all prepared in the same batch and had an identical history, so the different κ values were not readily explainable. Thickness was accounted for in the analysis, and measurements were performed on three locations per sample to account for local differences. Considering that the porosity should be the same (see Section 2.2), the differences between them must have been due to other minor process deviations that were not readily identifiable. The ageing also decreased the differences between samples, suggesting that the initial state of each sample may not have been equal.

The power factor (PF) and zT of the material were not determinable at any given point in time due to the different timing of each measurement. However, we could calculate approximate theoretical values at day 7 after synthesis by using the nearest datapoint for each variable. This gave a PF of $3.7\times 10^{-4}$ W m${}^{-1}$ K${}^{-1}$ and a zT of $4.2\times 10^{-4}$. Compared to other nanostructured n-type Bi_2_Te_3_, this is low, but the difference is predominantly caused by the greater than thousandfold lower electrical conductivity [24,29,42]. The zT increased slightly at day 30 due to an improvement in *S* and κ, even though this was largely negated by a further degradation of σ.

## 4. Discussion

Nanostructuring reduced the thermal conductivity of Bi_2_Te_3_ by one order of magnitude. Unfortunately, the electrical conductivity had been reduced by over two orders of magnitude and decreased further over time. This degradation can partly be ascribed to oxidation, which formed an insulating shell around the nanoparticles. The QCM measurements indicate that this could happen directly after synthesis, so, for an optimal synthesis process, more measures should be taken to prevent oxidation. Examples include a higher concentration of reducing gas (e.g., H_2_) [33], the addition of moisture and oxygen filters, or longer runtime before the deposition to purge the system [51]. Furthermore, it is crucial to protect these reactive nanoparticles after production. The results in this work suggest that a protective atmosphere can protect the deposits for the first few minutes, but, in the end, this only slows down the degradation. This can be expected for any nanoporous material, as air can penetrate deeply into the film and oxidise it throughout. Immediate further processing or packaging is thus required for nanoporous films to maintain their initial performance.

Considering these experiments were not aimed at producing the optimal material but at demonstrating the production method by using a common material, the Bi_2_Te_3_ performance can be improved in several ways. Firstly, the Bi_2_Te_3_ used in the experiments here was undoped, while it is known that doping or even alloying significantly improves the performance of TE materials [8,46]. Secondly, the porosity helped reduce the thermal conductivity, but was also detrimental to the electrical conductivity. Changing the deposition conditions or particle size to change the porosity is possible, but the film will remain highly porous. It is common to compress and sinter nanopowders, using spark plasma sintering (SPS), into pellets, to increase the electrical conductivity while maintaining the nanosized grains [48,52,53]. SPS requires larger quantities to enable pelletising, which this printing method cannot yet produce, but future versions of this equipment could. Compression by other means can be an alternative. By sandwiching the material between the two contacts of a thermopile, it can be simultaneously protected, packaged, and contacted. A roll-to-roll or flip-chip process could be developed with spark ablation and impaction printing to achieve this.

Lastly, this method can produce other TE materials. For example, producing oxides for high-temperature applications makes oxidation problems irrelevant. ZnO, a thermoelectric oxide, has been successfully deposited this way, and the first experiments have been conducted to assess its performance. Nanoparticle alloys that do not exist in a bulk variant can also be produced using two electrodes of different compositions [54].

## 5. Conclusions

In conclusion, the synthesis of TE materials using spark ablation is possible, and the fabrication of nanostructured TE devices was demonstrated using impaction printing. The one-step approach and direct writing capability make it a simple, clean, and versatile method for producing TE materials. The technology was demonstrated using undoped Bi_2_Te_3_ because it is the most common TE material. It exhibited n-type behaviour. The lowest found κ was 0.1 W m${}^{-1}$ K${}^{-1}$ at room temperature, comparable to state-of-the-art nanostructured Bi_2_Te_3_, but this came at the cost of the electrical conductivity. The material showed degradation over time, starting immediately after deposition: 37% reduction in conductivity was already observed within the first 25 minutes. This can be attributed to mild oxidation and adsorption of contaminants on the nanoparticle surface. To retain its performance, further protection of the film or immediate, air-tight packaging of the device is required. Although the method presented here can synthesise and deposit a wide range of TE materials without developing new synthesis processes, the process requires further optimisation. Its performance was lower than comparable nanostructured films, predominantly due to the low electrical conductivity. Improved material composition and synthesis parameters, perhaps combined with compression or annealing to reduce porosity, will improve the TE performance further.

## Figures and Tables

**Figure 1 nanomaterials-13-01778-f001:**
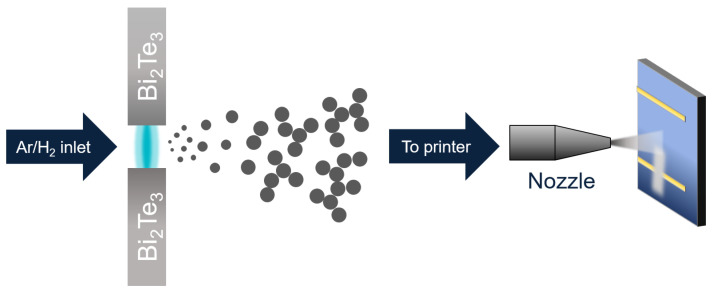
Diagram showing spark ablation (**left**) and impaction printing (**right**). An electric discharge ablates two electrodes, creating a metal vapour that condenses. The nanoparticles grow until they reach a final size, based on the synthesis conditions, and then agglomerate. A nozzle is placed close to a substrate, and the aerosol is printed on a substrate.

**Figure 2 nanomaterials-13-01778-f002:**
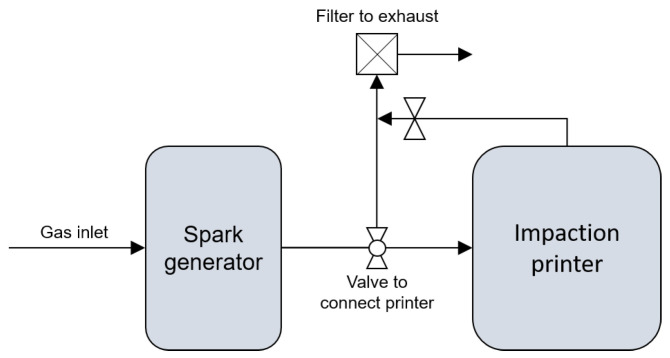
Schematic of the spark ablation and printer setup showing all gas inlets and outlets.

**Figure 3 nanomaterials-13-01778-f003:**
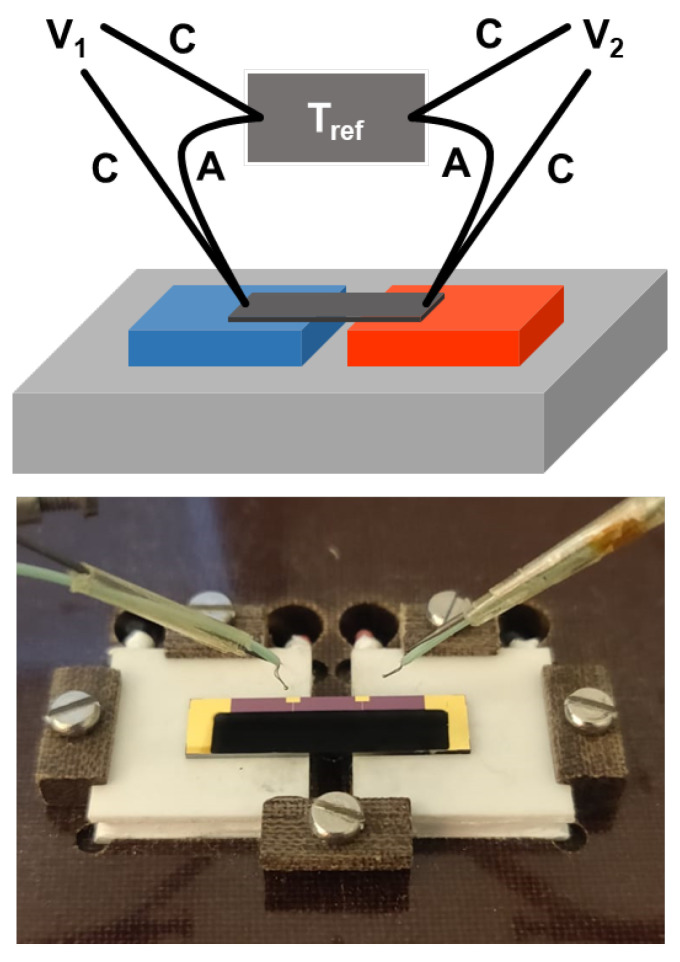
(**Top**): Schematic drawing of the Seebeck effect measurement setup. The device is placed on two Peltier elements to create a temperature gradient over the device. The voltage and temperature are read simultaneously using type K thermocouples at both ends of the device, and the setup contribution is subtracted from the acquired voltage. The thermocouples, made of chromel (C) and alumel (A), are used with a liquid nitrogen bath as the reference temperature. (**Bottom**): Picture of a device placed on the Peltier elements with the thermocouples (not in contact). The black strip is the Bi_2_Te_3_ film deposited on Au contacts.

**Figure 4 nanomaterials-13-01778-f004:**
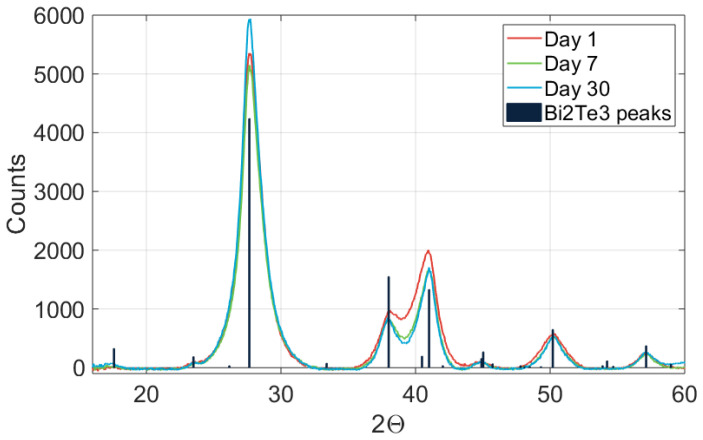
XRD spectrum showing a Bi_2_Te_3_ sample as freshly deposited (red), after one week (green), and after one month (blue).

**Figure 5 nanomaterials-13-01778-f005:**
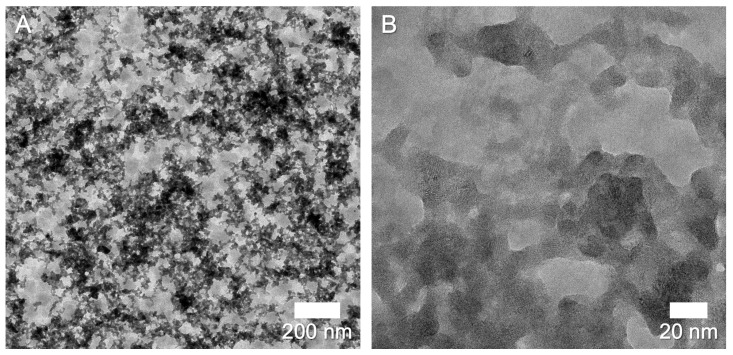
(**A**) TEM images of Bi_2_Te_3_ nanoparticles. (**B**) HR-TEM image. We can observe extensive agglomeration and neck formation between particles in an agglomerate. The background is blurry because the membranes have non-uniform transparency, due to processing imperfections during production and crystallinity of the AlO${}_{x}$ substrate. Printing speed was 200 mm min${}^{-1}$. All other settings were as mentioned before.

**Figure 6 nanomaterials-13-01778-f006:**
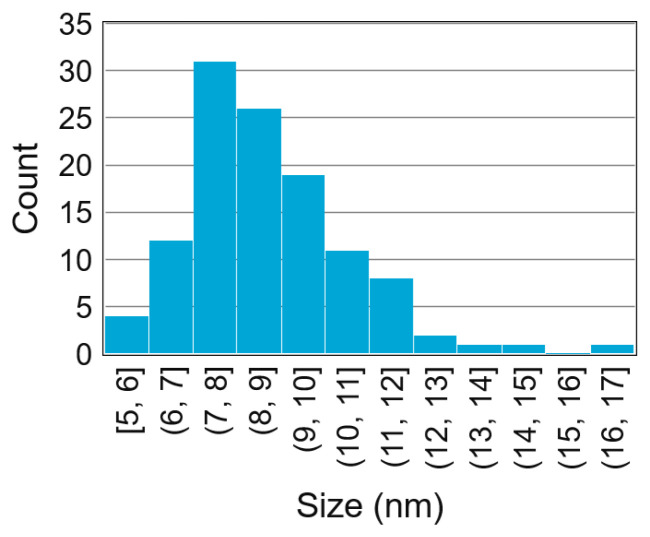
Size distribution of the diameter of 116 nanoparticles, measured by hand on 4 TEM images. Automated measurements were not possible due to the blurry background. The mean particle size was 8±2 nm.

**Figure 7 nanomaterials-13-01778-f007:**
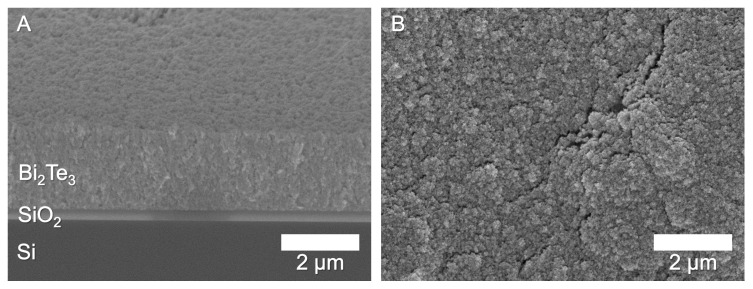
(**A**) Side view of a cleaved deposit with the SiO_2_ layer visible between the Si and Bi_2_Te_3_. The printed line is 2.1 μm thick. (**B**) Top view of the same sample, showing a rough surface.

**Figure 8 nanomaterials-13-01778-f008:**
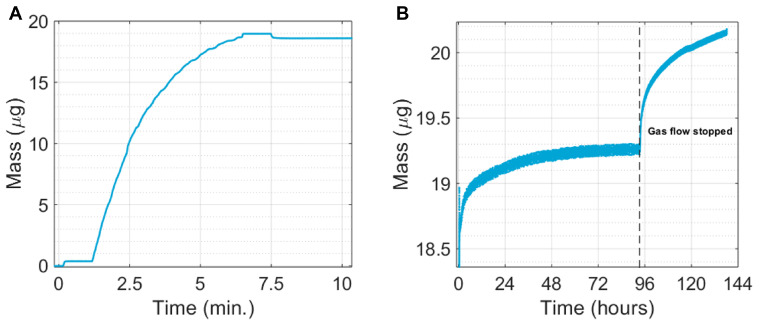
(**A**) Mass change over time during deposition, as recorded by a QCM. The first step resulted from the force exerted by the nozzle on the QCM. The decaying mass deposition rate was caused by the printing pattern and non-uniform response of the QCM. Once printing was finished, a step down happened when the nozzle left the printing area. The final deposited mass was 18.60 μg. (**B**) Mass change of the same sample after deposition. Initially, the Ar/H_2_ gas flow was still on, with the nozzle next to the QCM. Gas flow was closed off to see the effect of the deposition chamber in an “idle” state.

**Figure 9 nanomaterials-13-01778-f009:**
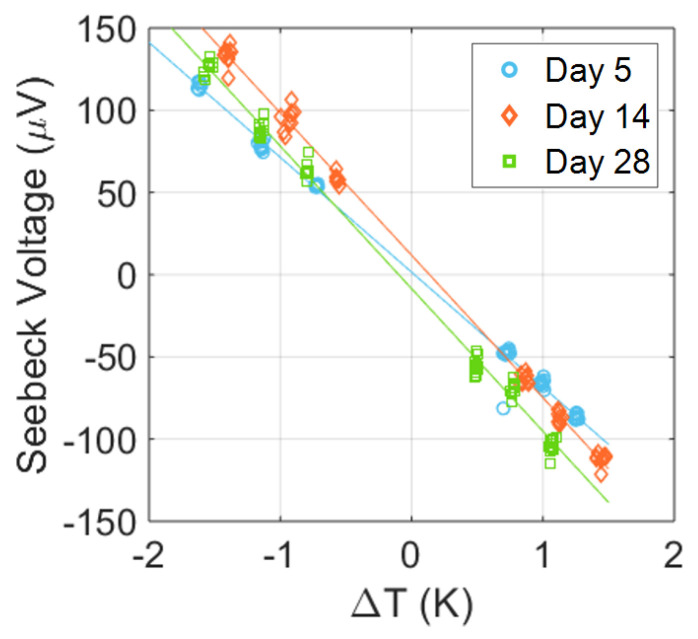
Seebeck voltage determined using the setup in Figure 3 on a sample measured at several points after synthesis. Repeated measurements were taken at room temperature with a ΔT created using Peltier elements. The slope of a least squares fit gave the Seebeck voltage. The first measurement had S=−88.3±1.2
μV K${}^{-1}$, which increased to S=−105.1±1.2
μV K${}^{-1}$ and did not increase further after two more weeks, reaching S=−105.2±1.6
μV K${}^{-1}$.

**Figure 10 nanomaterials-13-01778-f010:**
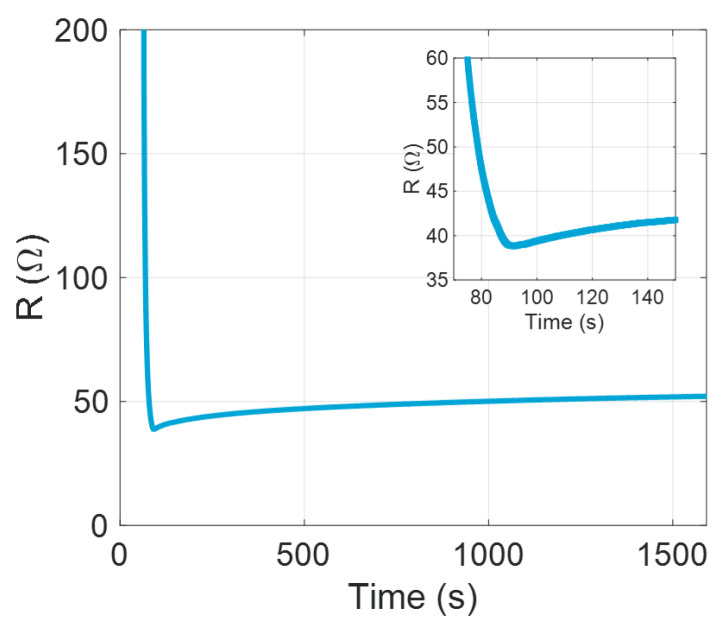
Resistance of a single wirebonded device during deposition. Until deposition, an open circuit is measured (not shown), after which a sharp decrease in the resistance is seen until the deposition area has passed entirely over the four electrodes and deposition ends, after which we see an increase due to oxidation and restructuring of the film. Inset: close-up at the deposition time. The lowest measured value was 38.82Ω. The film was kept under deposition conditions for the duration of the measurement.

**Figure 11 nanomaterials-13-01778-f011:**
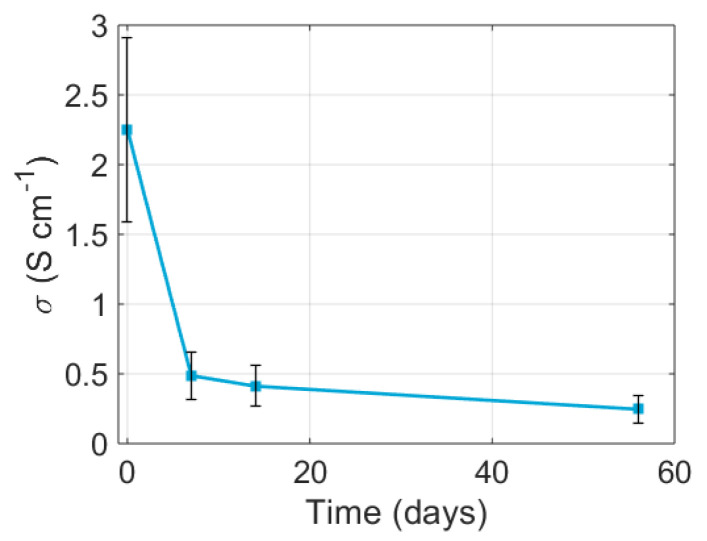
Decay of the mean conductivity of six deposited lines with 61 measurement points each. Measurements were performed within hours of deposition, so some degree of oxidation must be assumed at day zero. After a fast decrease by nearly a factor 5 in the first week, conductivity decay slowed down.

**Figure 12 nanomaterials-13-01778-f012:**
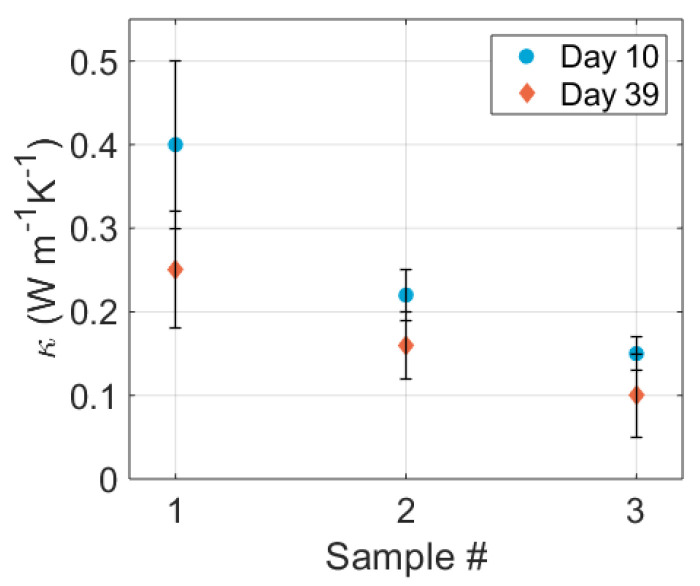
Thermal conductivity of three samples measured by LFA, measured 10 and 49 days after deposition. Each datapoint consisted of two measurements at different positions on the film, using a density of 44% that of bulk Bi_2_Te_3_: 3.40 g cm${}^{-3}$. A significant reduction in thermal conductivity can be observed for two out of three samples after ageing in air.

## Data Availability

All data presented here are available at the 4TU data repository: https://data.4tu.nl/institutions/Delft_University_of_Technology, (accessed on 15 May 2023).

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
