# Peer review of "Nanostructured Thermoelectric Films Synthesised by Spark Ablation and Their Oxidation Behaviour"

_nanomaterials, 2023, doi:10.3390/nano13111778_

Round 1
Reviewer 1 Report
This paper is well-written and shows a novel approach and useful results. Some comments are given for authors to improve their work.
1. Although authors have provided a reference, it would be useful to describe the method for measuring the porosity of the thin film from QCM and the uncertainty of this approach.
2. Similarly, it would be useful to describe the method for determining the thermal conductivity of the thin film from the bidirectional heat flow approach and the uncertainty of this method.
3. Since the porosity is one of the major factor on the thermal conductivity, it is suggested to compare the porosity of three samples (Sample #1, #2, and #3 shown in Figure 12).
4. Authors are suggested to explain why the sample (S3) with a lower spark voltage shows a lower thermal conductivity.
5. What are the reasons causing the decrease of thermal conductivity with increasing the time after deposition, as shown in Fig. 12?
Author Response
Thank you for your time to review our manuscript and your valuable comments. You can find our responses to your comments below, addressed per point.
- Although authors have provided a reference, it would be useful to describe the method for measuring the porosity of the thin film from QCM and the uncertainty of this approach.
The corresponding section has been expanded to provide a more detailed description.
- Similarly, it would be useful to describe the method for determining the thermal conductivity of the thin film from the bidirectional heat flow approach and the uncertainty of this method.
This section has also been expanded too.
- Since the porosity is one of the major factor on the thermal conductivity, it is suggested to compare the porosity of three samples (Sample #1, #2, and #3 shown in Figure 12).
The QCM method does not measure the porosity of the samples directly, but uses two other depositions (a line for the cross-sectional area and one on the QCM for deposition rate) to determine the porosity of a deposition setting. The porosity was determined using three samples, each measured three times. The referenced paper (10.1088/1361-6528/ac7811), showed that for gold, the density is independent of the spark ablation or deposition settings for the conditions tested. It is assumed this is valid for Bi2Te3, so the three samples are assumed to have identical porosities.
This is clarified in the manuscript now.
- Authors are suggested to explain why the sample (S3) with a lower spark voltage shows a lower thermal conductivity.
A discussion of this is added. The three samples show a wide range of values, so it is unsure if the lower spark voltage caused this difference. A lower voltage does create a smaller particle size distribution, but it is difficult to quantify this effect without a larger sample size.
- What are the reasons causing the decrease of thermal conductivity with increasing the time after deposition, as shown in Fig. 12?
We contribute this to degradation of the films by oxidation. Oxidation is slow, but happens, according to Bando et al (10.1088/0953-8984/12/26/307). A (thicker) oxide shell around the nanoparticles will add an insulating layer, reducing the thermal conductivity (and reducing the electrical conductivity). Unfortunately, there has not been a systematic study of the thermal conductivity of the Bi2Te3-oxide system, so it is not possible to quantify this effect.
Reviewer 2 Report
In the paper of “Nanostructured thermoelectric films synthesized by spark ablation and their oxidation behaviour”, authors developed a new method based on spark ablation nanoparticle generation to create nanostructured thermoelectric materials. This is a topic of interest to researchers in related fields. Overall, the paper could be accepted for publishing after revisions.
1. Generally, extrinsic hole or electron doping introduces additional antisite defects, which shift the Fermi level into the valence or conduction band, resulting in the p- or n-type Bi2Te3 with an optimized TE performance. In this work, the n-type of Bi2Te3 was obtained by excessive oxidation or by organic compounds? It is helpful for readers if provide more explicit evidence and coexist with proper references.
2. In parallel, it is suggested that the authors add the discussion on the relationship (Seebeck coefficient vs days) between Seebeck coefficient and samples in the section of 3.2, so that the significant variation of Seebeck coefficient in can be emphasized.
3. Using the electrical conductivity and Seebeck coefficient, the power factor and zT can be calculated and shown in a plot with some discussion and comparison with previous studies.
4. Conclusion needs to be enhanced by adding the obtained results.
Author Response
Thank you for your time to review our manuscript and your valuable comments. You can find our responses to your comments below, addressed per point.
- Generally, extrinsic hole or electron doping introduces additional antisite defects, which shift the Fermi level into the valence or conduction band, resulting in the p- or n-type Bi2Te3 with an optimized TE performance. In this work, the n-type of Bi2Te3was obtained by excessive oxidation or by organic compounds? It is helpful for readers if provide more explicit evidence and coexist with proper references.
In this work, there were no organic materials used at any point of the synthesis process, so it is unlikely any organic contaminants have acted as dopant. It is possible oxygen acts as an extrinsic dopant, but nanostructured Bi2Te3 seems to exhibit predominantly n-type behavior. These references have been added to expand on it in the text. (https://doi.org/10.1016/j.cej.2020.124295, https://doi.org/10.1021/acsami.9b13920, https://doi.org/10.1021/acsami.9b12079,https://doi.org/10.1016/j.apsusc.2017.02.187, https://doi.org/10.1002/aelm.201800904)
- In parallel, it is suggested that the authors add the discussion on the relationship (Seebeck coefficient vs days) between Seebeck coefficient and samples in the section of 3.2, so that the significant variation of Seebeck coefficient in can be emphasized.
The ageing of the sample changes the composition of the film slowly, so the charge carriers and mobility will change and conductivity are all affected. It is without measuring all of these parameters difficult to say what the reason is for a higher Seebeck effect after aging. Figure 9 shows only one sample, measured several times on the same spot per day. The section has been expanded to discuss the change better.
- Using the electrical conductivity and Seebeck coefficient, the power factor and zTcan be calculated and shown in a plot with some discussion and comparison with previous studies.
A theoretical zT has been calculated and is presented in section 3.2 and discussed. However, plotting its change over time is not possible because the measurements do not all have the same time intervals. Because of the different times of measurement, there is no zT for any time instant, and it is therefore not representative to express the film performance.
- Conclusion needs to be enhanced by adding the obtained results.
Changed accordingly.